# An Identification System Targeting the SRK Gene for Selecting S-Haplotypes and Self-Compatible Lines in Cabbage

**DOI:** 10.3390/plants11101372

**Published:** 2022-05-21

**Authors:** Wendi Chen, Bin Zhang, Wenjing Ren, Li Chen, Zhiyuan Fang, Limei Yang, Mu Zhuang, Honghao Lv, Yong Wang, Yangyong Zhang

**Affiliations:** Key Laboratory of Biology and Genetic Improvement of Horticultural Crops, Institute of Vegetables and Flowers, Chinese Academy of Agricultural Sciences, Ministry of Agriculture, Beijing 100081, China; chenwd9611@163.com (W.C.); 13126720352@163.com (B.Z.); 17863805323@163.com (W.R.); 18205480752@163.com (L.C.); fangzhiyuan@caas.cn (Z.F.); yanglimei@caas.cn (L.Y.); zhuangmu@caas.cn (M.Z.); lvhonghao@caas.cn (H.L.); wangyong@caas.cn (Y.W.)

**Keywords:** cabbage, self-incompatibility, S-haplotype, primer set, cross-incompatibility

## Abstract

Cabbage (*Brassica oleracea* L. var. *capitata*) self-incompatibility is important for heterosis. However, the seed production of elite hybrid cannot be facilitated by honey bees due to the cross-incompatibility of the two parents. In this study, the self-compatibility of 58 winter cabbage inbred lines was identified by open-flower self-pollination (OS) and molecular techniques. Based on the NCBI database, a new class I S-haplotype-specific marker, PKC6F/PKC6R, was developed. Verification analyses revealed 9 different S-haplotypes in the 58 cabbage inbred lines; of these lines, 46 and 12 belonged to class I (S6, S7, S12, S14, S33, S45, S51, S68) and class II (S15) S-haplotypes, respectively. The coincidence rate between the self-compatibility index and S-haplotype was 91%. This study developed a Tri-Primer-PCR amplification method to rapidly select plants with specific S-haplotypes in biased segregated S-locus populations. Furthermore, it established an S-haplotype identification system based on these nine S-haplotypes. To overcome parental cross-incompatibility (18-503 and 18-512), an inbred line (18-2169) with the S15 haplotype was selected from the sister lines of self-incompatible 18-512 (S68, class I S-haplotype). The inbred line (18-2169) showed self-compatibility and cross-compatibility with 18-503. This study provides guidance for self-compatibility breeding in cabbage and predicts parental cross-incompatibility in elite combinations.

## 1. Introduction

Cabbage (*Brassica oleracea* L. var. *capitata* L.) is an important vegetable crop in the Brassicaceae family and is widely cultivated throughout China. A self-incompatibility (SI) system is a mechanism that evolved through natural selection to prevent degradation caused by self-pollination, which is beneficial for maintaining the heterozygosity of individual genes and improving the survival and competition ability of species [1,2]. SI is currently found in over half of flowering plants, involving 70 families and 250 genera [3]. Cabbage belongs to the sporophytic self-incompatibility system, mainly controlled by a single, multiallelic locus, the S locus, which contains the genes encoding the female and male S determinants [4,5]. There are many alleles at the S locus, also known as S-haplotypes. Within this locus, three genes have been identified, including SLG (S-locus glycoprotein), SRK (S-locus receptor kinase), and SCR/SP11 (S-locus cysteine-rich protein and S-locus protein 11) [5,6,7,8,9].

The SLG and SRK genes are expressed in the stigma, and the SCR/SP11 gene is expressed in the anther. These three genes play an important role in the recognition and response mechanisms between pollen and stigma [10,11]. The SRK gene is the sole determinant of the specificity of stigma S, and the SLG gene has been shown to enhance the recognition process between self-pollen and stigma [12]. S-haplotypes in Brassica have been classified into two classes, class I and class II: class I S-haplotypes possess strong self-incompatibility and class II S-haplotypes possess weak self-incompatibility, respectively [5]. To date, over 50 S-haplotypes have been found in *B. oleracea*, and over 20 S-haplotypes have been found in cabbage [13].

Cabbage hybrids are generally developed with strong heterosis and excellent characteristics (for example, disease resistance and cold tolerance). Self-incompatibility and male sterility are two main methods to exploit heterosis. From the 1970s to the 1990s, almost all cabbage hybrids were made with self-incompatible lines. However, there were some defects in the method of self-incompatibility, such as an inability of the hybrid rate to reach 100% and the high cost of the artificial self-pollination of parents in the bud stage. After the discovery and creation of male-sterile resources, male sterility has been widely applied in cabbage breeding programs over the last 20 years. With the wide usage of male sterility, self-compatible lines are favored over self-incompatible lines [14]. Self-compatible lines can be self-pollinated by honey bees to prevent artificial pollination. Therefore, the accurate and quick selection of self-compatible lines is urgently needed.

In the past, the most common method to identify S-haplotypes was to conduct flower pollination with different S-haplotype standard lines. If the two materials contained the same S-haplotype, they would be incompatible (compatible index < 1.0). If two materials had different S-haplotypes, they would be compatible. However, due to the large number of S-haplotypes of cabbage, this method was time-consuming and labor-intensive. Moreover, the breeding of self-incompatible individuals ignores or eliminates self-compatible haplotypes. Therefore, it is necessary to find a quick and convenient way to identify and analyze S-haplotypes in early generations and enrich S-haplotypes in cabbage varieties, especially self-compatible S-haplotypes.

With the rapid development of molecular marker technology and the functional identification of the S locus, molecular markers are increasingly used for the identification of S haplotypes [15,16,17]. In Brassica, two pairs of primers, PK1/PK4 and KD4/KD7, have been designed to distinguish class I S-haplotypes and class II S-haplotypes, respectively. To specifically amplify the SRK gene, T. Nishio et al. (1994) designed several primers and finally obtained the optimal primer PK1/PK4. A portion of the class I S-haplotypes can be amplified using this pair of primers. Furthermore, J. I. Park et al. (2002) designed primers based on the SRK nucleotide sequences (kinase domain) of S2, S5 and S15, which belong to the self-compatible group of class II S-haplotypes. However, it was found that the bands amplified by PK1/PK4 were unstable and that the primers were poorly versatile in our previous experiments. In addition, recent studies have revealed that the separation ratio of S-haplotypes deviates. In a 2021 study of self-compatibility mutants in potatoes, it was found that a separation ratio of AA:Aa = 1:1 was produced by the potato of the Aa genotype after self-pollination for one generation instead of the expected AA:Aa:aa = 1:2:1 ratio [18]. Furthermore, in pummelo, the S-haplotypes of the progeny from the self-pollinated ‘GY’, whose S-haplotype was S1S2, segregated into only two classes (S1S2:S2S2) at a distorted ratio of 274:256. This ratio was consistent with an abnormity linked to the S2 locus [19]. Interestingly, we also found the phenomenon of S-haplotype abnormal separation in cabbage and developed two sets of specific primers to solve the problem of loss of S-haplotypes.

An elite winter cabbage hybrid, ‘Zhong Gan 2418’, was recently developed. However, hybrid seeds cannot be produced by honey bees due to the cross-incompatibility between the parent lines (18-503 and 18-512). To solve this problem, this study aimed to design a novel primer to distinguish the S-haplotype and use this primer to assist in selecting self-compatible lines, which could provide important guidance for cabbage self-compatible inbred line selection and hybrid development.

## 2. Results

### 2.1. Self-Pollination and Identification of Self-Compatible Index

The seed number was counted after flower pollination, and the self-compatibility index was calculated, revealing 12 self-compatible lines (Figure 1A) and 46 self-incompatible lines (Figure 1B). Among the self-compatible lines, the proportions of early-maturation, medium-maturation and late-maturation cabbage were 11%, 44% and 19%, respectively. Flat-headed lines and round-headed lines are two common head types. The ratio of width to height is close to 1:1 in round-headed lines, and the ratio of height to width is less than 0.8:1 in flat-headed lines. Among the flat-headed lines, 17.24% were compatible, and all of them originated from Japan. In contrast, among the Japanese round-headed lines, all except for 18-605 showed self-incompatibility. Among the round-headed lines overall, 24.13% were compatible, and all of them originated from the Netherlands except 18-605. Therefore, most of the self-compatible flat-headed lines originated from Japan, and most of the self-compatible round-headed lines originated from the Netherlands.

### 2.2. Marker Development for S-Haplotype Identification

While using primer PK1/PK4 to distinguish the S-haplotypes of 58 cabbage materials, the most substantial problem was poor versatility and repeatability (only three materials were successfully amplified). The similarity between the PK1/PK4 primer sequences and S-haplotype sequences downloaded from the NCBI database was analyzed (Table 1). Among the 29 known S-haplotypes, the PK1/PK4 primer matched the SRK sequences of only 13 S-haplotypes, with over 90% similarity. There was poor matching among the remaining 16 S-haplotypes (less than 90% similarity). We inferred that PK1/PK4 was suitable for less than half of the S-haplotypes in cabbage; thus, it was necessary to develop new primers capable of amplifying more class I S-haplotypes. The matching degree of PKC6F/PKC6R among all class I S-haplotypes was 89%, much higher than that of PK1/PK4. The sequence alignment results indicate that the primers PKC6F/PKC6R can amplify almost all known S-haplotypes (except S17 and S63). In addition, no band was amplified from class II haplotypes by the new PKC6F/PKC6R primer pair.

### 2.3. Identification of S-Haplotypes of 58 Cabbage Inbred Lines

According to the NCBI sequences of S-haplotypes, the primer pair PKC6F/PKC6R was expected to amplify approximately 480 bp bands from class I S-haplotype materials (Figure 2); the primer pair KD4/KD7 was expected to amplify approximately 1100 bp bands from class II S-haplotype materials (Figure 3). The bands amplified by the two pairs of primers from 58 materials (Appendix A) revealed that 15 of the 58 materials were class II S-haplotypes and 43 were class I S-haplotypes.

By sequencing and BLAST analysis of the amplicons from primers PKC6F/PKC6R and KD4/KD7, the S-haplotype of each inbred line was determined. The 58 inbred lines were classified into 9 S-haplotypes, including 8 class I S-haplotypes (S6, S7, S12, S14, S33, S45, S51, S68) and only one class II S-haplotype (S15). S15 had the highest frequency among the S-haplotypes (26.6%), followed by S68 (21.9%). The coincidence rate between the self-compatibility index and S-haplotype was 91%. Of note, 18-512 and 18-503 shared the same S-haplotype (S68), explaining why the parents of the winter cabbage hybrid ‘Zhonggan 2418’ could not produce seeds during cross pollination in the flower stage.

There were six S-haplotypes in both the flat-headed and round-headed lines, but the type of S-haplotype substantially differed between these lines. Only S15 and S68 S-haplotypes are shared by both sets of lines. The unique S-haplotypes in the round-headed lines are S7, S14, S33 and S45. S6, S12 and S51 are the unique S-haplotypes in the flat-headed lines. As the distribution of S-haplotypes differed substantially based on head type, enrichment of the S-haplotypes in different head types should be performed for cabbage breeding.

### 2.4. Self-Compatible Line Identification among the Sister Lines of 18-512

Since 18-512 shared the same S-haplotype as 18-503, the elite hybrid ‘Zhonggan 2418’ could not conduct the flower cross-pollination by honey-bees. In this study, we used the markers PKC6F/PKC6R and KD4/KD7 to select materials with class II S-haplotypes among the 18-512 sister lines and used these lines to overcome parental cross-incompatibility. The pedigree of the sister lines was shown in Appendix A. 18-512 and its sister line 18-2170 both have class I S-haplotypes, but its sister line 18-2169 has a class II S-haplotype (Figure 4 and Figure 5). Field pollination at the flowering stage also confirmed the accuracy of molecular identification (Appendix A; Figure 6). The results showed that 18-512 and 18-2170 showed self-incompatibility at the flowering stage, while 18-2169 was self-compatible. In addition, 18-503 and 18-2169 showed cross-compatibility, while 18-503 and 18-2170 showed incompatibility (Figure 7). Sequencing and comparison were carried out on the amplified products of PKC6 primers, and it was determined that all the sister lines of 18-512 were S68 except for 18-2169, which was S15.

Based on the results of previous studies [20], we selected 18 pairs of SSR primers (Appendix A) to explore the genetic background similarity among 18-512, 18-2169 and 18-2170. The results showed that the genetic backgrounds of these three lines were similar (Figure 8); thus, 18-2169 could replace 18-512 to produce seeds, and the elite winter cabbage hybrid ‘Zhonggan 2418’ was successfully pollinated by honey bees in the spring of 2021. It was found that there was no significant difference in head weight, height, width and biological yield when the self-compatible line 18-2169 or self-incompatible line 18-512 were crossed with the same female parent 18-503.

### 2.5. Development of the Class I S-Haplotype Primer Identification System

The sequences of S7/S33 (C06: 32,096,418–32,097,017) and S33/S45 (C06: 32,097,589–32,097,017) in TO1000 (http://plants.ensembl.org/Brassica_oleracea/Info/Index, accessed on 25 June 2021) were compared to design specific primers. The length of amplified bands from each S-haplotype was predicted (Table 2). Newly developed primers were tested with known S-haplotype materials (Figure 9). According to the results, we found that primers TS33-F/TS7-F/TS-R and TF33-F/TF45-F/TS-R were the optimal primer sets to distinguish S7/S33 and S33/S45, respectively. The two sets of codominant primers could be used to directly distinguish the three S-haplotypes of *B. oleracea* by one-step Tri-Primer-PCR without sequencing, which is a more convenient and rapid method for the differentiation of the S-haplotypes in *B. oleracea*.

The S-haplotypes of cabbage ‘1186’ and ‘emerald’ was identified as S7S33 and S45S33, respectively, by PKC6-F/R and PCR product sequencing. Further identification of their self-pollinated progenies showed that the separation ratio was found to be biased from the normal ratio (1:2:1) after self-pollination. The actual separation ratio is S7S7:S7S33:S33S33 = 4:26:27 and S45S45:S45S33:S33S33 = 3:32:29, respectively. Both S7S33 and S45S33 show a tendency to deviate from S33. Therefore, S33 has a greater genetic advantage than S45 and S7. S45 and S7 are more likely to be lost without selecting. The partial segregation of dominant genes also explained the high proportion of S33 in class I S-haplotypes among the 58 wintering cabbage materials identified in this study. Moreover, in the process of separation and breeding of the progeny of cabbage hybrid ‘1186’ and ‘emerald’, we found that after more than five generations of self-pollination, most of the inbred lines were S33. Therefore, to protect the diversity of S-haplotypes in cabbage, it is necessary to develop markers to retain vulnerable S-haplotypes quickly and accurately.

Based on the characteristic sequences of different S-haplotypes, a class I S-haplotype classification system that uses three pairs of markers and three enzyme digestions was established; this system can be used to classify the class I S-haplotypes of the wintering cabbage commonly used in breeding. After PKC6 primers were used for amplification to distinguish the class I S-haplotypes, *Nal* IV was used for the first digestion, and 8 S-haplotypes were divided into 4 groups (Figure 10): S33 and S68 (Group 1, 480 bp); S12, S14 and S51 (Group 2, 432 bp + 48 bp); S7 and S45 (Group 3, 315 bp + 118 bp); and S6 (Group 4, 268 bp + 165 bp + 48 bp). In Group 1, the marker ID was designed based on the 6 bp length difference between S33 and S68 to realize one-step identification of S33 and S68 S-haplotypes without sequencing. In Group 2, the identification results of the three S-haplotypes were obtained by two enzyme digestions. *B**ts* I was used for the first digestion; this enzyme was unable to cleave S12, but could cleave S14 and S51 to 280 bp and 200 bp. S14 and S51 could be further distinguished by the second digestion with *P**ts* I. this enzyme was unable to cleave S51 but cleaved S14 into 314 bp and 166 bp pieces. In Group 3, S7 and S45 were distinguished by using each of the two pairs of primers developed above, namely TS7-F/TS-R and TF45-F/TS-R (Figure 11). Through the S-haplotype identification system, the 8 class I S-haplotypes in this study could be quickly distinguished without sequencing comparison analysis.

## 3. Discussion

### 3.1. Consistency of the Molecular Identification and Compatible Index (CI)

For most of the inbred lines, the molecular identification was consistent with the compatibility index, i.e., self-compatible with class II S-haplotypes and self-incompatible with class I S-haplotypes; the coincidence rate was 91%. There are several reasons for the remaining 9% anticoincidence. (i) Self-incompatibility is controlled by several loci, and SRK is the major, but not the only gene controlling self-incompatibility. The following three genes have been found to be linked to the locus associated with self-incompatibility: SLG (S-locus glycoprotein), SRK (S-locus receptor kinase), and SCR/SP11 (S-locus cysteine-rich protein and S-locus protein 11) [5,7]. In addition, some regulators that are not associated with the S locus, such as ARC1, THL1/THL2, and MLPK, are also involved in pathways related to self-incompatibility reactions [21,22,23,24,25]. (ii) The compatibility index is easily affected by environmental conditions, including temperature, humidity and pollination date. For the five lines with inconsistency between CI and S-haplotypes, we will construct a segregation population to map the novel genes associated with self-compatibility and self-incompatibility.

### 3.2. Distribution of S-Haplotypes in Winter Cabbage Materials

Among the 58 winter cabbages analyzed, 43 inbred lines belonged to the class I S-haplotype, and 15 inbred lines belonged to the class II S-haplotype. In this study, the occurrence frequency of different S-haplotypes in cabbage materials ranged from 3.45% to 25.86% (Appendix A), among which S15 had the highest occurrence frequency, accounting for 25.86%. S68 followed, accounting for 24.14%. S6 had the lowest frequencies (3.45%). This showed that the haplotypes did not exhibit an equal distribution and that the S-haplotype distribution was biased. Furthermore, the abnormal distribution was also confirmed by two F_2_ populations, generated from the F_1_ with S7S33 and S33S45 haplotype.

Self-incompatible lines and male-sterile lines are two important tools for utilizing cabbage heterosis [26]. To date, our research group has developed several self-incompatible lines with elite characteristics and cultivated F_1_ hybrids, such as ‘Zhonggan 11’ and ‘Zhonggan 23’, by using self-incompatible lines. Due to the long-term use of self-incompatible line breeding, it is easy to neglect class II S-haplotypes. Currently, more than 20 S-haplotypes have been discovered in cabbage, but only 9 S-haplotypes were found among the 58 winter cabbages analyzed. This result indicates that the diversity of the S-haplotype is not rich in the current breeding materials of winter cabbage and that it is urgent to introduce and utilize a wider range of S-haplotype cabbage varieties to prevent the cross-incompatibility of hybrid combinations. Notably, in this study, there were 15 self-compatible materials with the S15 haplotype. These results are consistent with those of previous studies in which S15 was found to be widely distributed in cabbages [27] and associated with weaker self-incompatibility [28]. In addition, we speculate that S15 might be the dominant S haplotype in winter cabbage.

### 3.3. Comparison of Different S-Haplotypes Identification Methods

The compatibility index method is widely used to identify S-haplotypes. Usually, the plants are cross-pollinated with the standard S-haplotype test lines, and the S-haplotype is determined by calculating the compatibility index [29]. This method is time-consuming and is susceptible to environmental changes. Fluorescence microscopy uses water-soluble aniline blue to show the elongation of pollen tubes in the stigma [30]. In 2014, a simple, inexpensive, rapid and semiquantitative method for the measurement of pollen tube growth in microtiter plates was published. This method relies on Congo Red binding to pollen tubes and correlates dye fluorescence to tube length [31]. With the rapid development of molecular marker technology, an increasing number of researchers have used molecular markers to identify the S-haplotype at the seedling stage. In previous studies, conservative primers were used for amplification, and PCR-RFLP markers were used to identify the S-haplotype in Brassica [15,32,33,34,35]. However, this method is not applicable for distinguishing S-haplotypes with no RFLP enzyme polymorphism.

At present, most S-haplotype identification of cabbage relies on the amplification of SRK sequences and sequence alignment. This method has certain advantages: (1) Compared with the previous field hybridization method, the sequencing comparison method reduces the consumption of manpower; and (2) sequencing comparison can better find some SNP mutation sites, making the identification of S-haplotypes more accurate and allowing the discovery of new S-haplotypes. In this study, a partial S-haplotype identification system was established; this system can directly select 9 S-haplotypes in winter cabbage breeding without sequencing. Hopefully, we will enrich the S-haplotypes of winter cabbage in the future, at that time we will continue to strive to establish a more complete and accurate S-haplotype primer identification system. If it can be realized, this method will provide a feasible method for the rapid identification of the S-haplotype and the utilization of cabbage breeding.

## 4. Materials and Methods

### 4.1. Plant Materials

In this case, 58 winter cabbage high-generation inbred lines (Appendix A) and sister lines of 18-512 were provided by the Institute of Vegetables and Flowers, Chinese Academy of Agricultural Sciences.

### 4.2. Identification of Self-Compatible Index

All flower self-pollination was performed by hand at approximately 9:00 a.m. and 3:00 p.m. Approximately 60 days after self-pollination of open flowers, the seed pod number and seed particle number of each plant were investigated with three repetitions, in which 10 buds were analyzed per repetition. The compatibility index (CI) was calculated as follows: Self-compatibility index = $\frac{\mathrm{number}\;\mathrm{of}\;\mathrm{seeds}}{\mathrm{number}\;\mathrm{of}\;\mathrm{pods}\;\mathrm{of}\;\mathrm{pollinated}\;\mathrm{flowers}}$ (self-incompatibility: CI < 1; self-compatibility: CI ≥ 1).

### 4.3. Novel Marker Development for Class I S-Haplotype Identification

The primer pairs PK1/PK4 [33] and KD4/KD7 [20] were designed to amplify class I S-haplotype materials and class II S-haplotype materials, respectively. However, while PK1/PK4 was used in this study to determine whether it could be used to select class I S-haplotypes, the amplified bands of PK1/PK4 were unstable and poorly versatile. Therefore, over 20 cabbage S-haplotype DNA sequences downloaded from the NCBI GenBank database (GenBank accessions are shown in Table 1) were aligned and PKC6 (Table 3), a novel allele-specific SRK primer pair, was developed to amplify class I S-haplotypes. The primer pair was designed using Primer Premier 5 according to the conserved region sequence of the S domain of SRK (Figure 12).

### 4.4. Identification of S-Haplotypes of 58 Cabbage Inbred Lines

The basic method of DNA extraction (CTAB) was improved, and genomic DNA was extracted from leaf tissue [36]. The DNA concentration was determined by a NanodropNd-100 Spectrophotometer, and the DNA was diluted to a working concentration of 40–100 ng/μL. DNA quality was determined by 1.5% agarose gel electrophoresis, and the DNA was stored at 4 °C.

Using genomic DNA as templates, PKC6F/PKC6R and KD4/KD7 were used for PCR amplification. The 20 μL PCR mixture contained 4 μL of DNA template, 2 μL of 10× PCR buffer (Mg^2+^ included), 1.6 μL of dNTPs, 1 μL of the forward primer, 1 μL of the reverse primer, 0.4 μL of Taq DNA polymerase (2.5 U/μL) and 10 μL of ddH2O. Reactions were performed in a thermal cycler as follows: 94 °C for 5 min; 33 cycles of 94 °C for 30 s, 55 °C for 30 s and 72 °C for 45 s; and 72 °C for 5 min. PCR amplification products were separated in 1.2% agarose gels in 1× TBE buffer and visualized under UV light.

PCR products were purified and sequenced by Shanghai Majorbio Bio-pharm Technology Co., Ltd. (Shanghai, China). The sequences were aligned in NCBI using BLAST, and the S-haplotypes of 58 winter cabbage materials were determined.

### 4.5. Self-Compatible Line Screening among the Sister Lines of 18-512

An elite winter cabbage hybrid, ‘Zhong Gan 2418’, with two parents, 18-503 and 18-512, was developed. To test the cross-compatibility of this hybrid, two parents were crossed with each other during the open flower stage. If cross-incompatibility existed, sister lines of 18-512 were used to screen for self-compatible lines with the class II S-haplotype. In total, five sister lines, including 18-512, 18-2169, 18-2170, 19-466, 19-467 and 19-468, were screened by two pairs of primers, PKC6F/PKC6R and KD4/KD7. Additionally, the seed-setting rates of open flower self-pollination and open flower cross-pollination of these sister plants with 18-503 were determined (Appendix A).

To compare the genetic background of 2169 and other plants of the 18-512 sister line, we selected several pairs of cabbage fingerprint markers for verification [31,37]. The PCR method used was the same as that described in Section 2.4.

### 4.6. Development of the Class I S-Haplotype Primer Identification System

Although PKC6F/PKC6R and KD4/KD7 can be used for the identification of classes I and II, certain S haplotypes still cannot be determined by PCR alone. Therefore, in this study, a new primer combination, named Tri-Primer-PCR, was developed to directly identify some S-haplotypes. Specific primers were designed to distinguish two pairs of S-haplotypes (S7/S33 and S33/S45) according to the sequences published on NCBI and the sequencing results in this study. Two hybrid varieties (‘1186’ is S7/S33 and ‘emerald ’ is S33/S45) and their derived inbred lines. 

Conditions and procedures of enzyme digestion. The following reaction was set up on ice: 1 μg of DNA, 5 μL of 10× NEBuffer, 1 μL of 10 units restriction enzyme, and nuclease-free water to 50 μL. The reaction was gently mixed by pipetting up and down and briefly microfuged. It was incubated at 37 °C for 60 min and then heat inactivated at 65 °C for 20 min.

## Figures and Tables

**Figure 1 plants-11-01372-f001:**
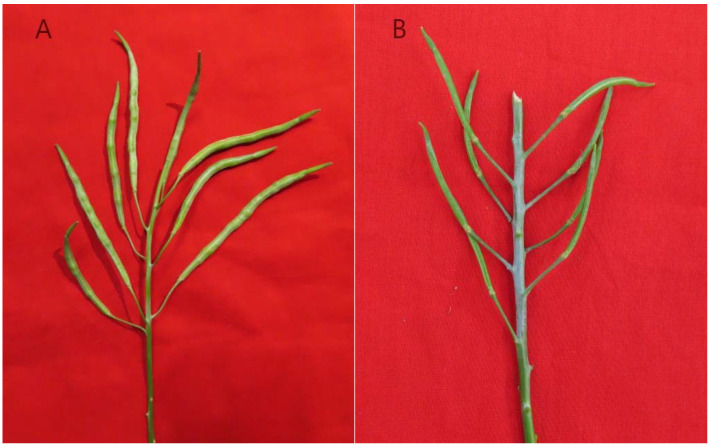
Pod enlargement of self-compatibility lines 18-493 (**A**) and self-incompatibility lines 18-551 (**B**) in the open flower stage.

**Figure 2 plants-11-01372-f002:**
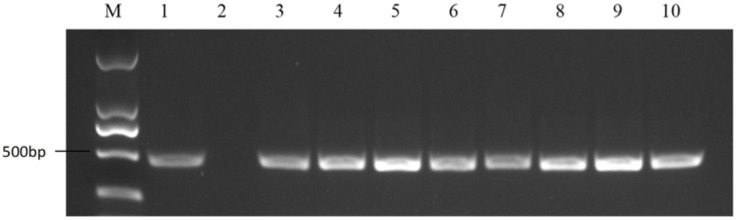
PCR amplification of PKC6F/PKC6R in cabbages. M represents the DNA Marker 2000; 1 represents the positive control; and 2 represents the negative control. Lanes 3–10 are cabbage materials with class I S-haplotypes.

**Figure 3 plants-11-01372-f003:**
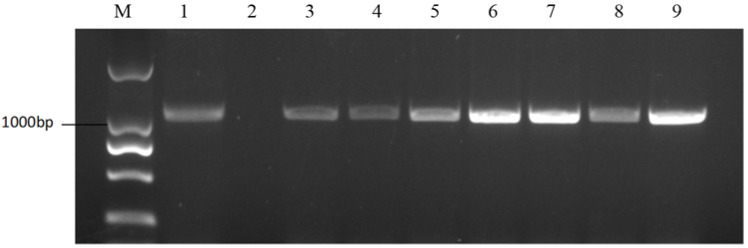
PCR amplification of KD4/KD7 in cabbages. M represents the DNA Marker 2000; 1 represents the positive control; and 2 represents the negative control. Lanes 3–9 are cabbage materials with class II S-haplotypes.

**Figure 4 plants-11-01372-f004:**
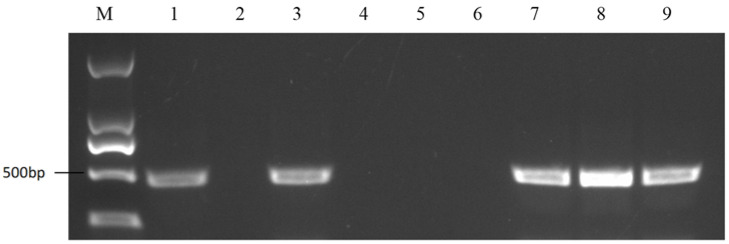
PCR amplification of PKC6F/PKC6R in 18-512 sister lines. M represents the DNA Marker 2000; 1 represents the positive control; and 2 represents the negative control. Lanes 3–9 are 18-512, 18-2169-1, 18-2169-2, 18-2169-3, 18-2170-1, 18-2170-2, and 18-2170-3.

**Figure 5 plants-11-01372-f005:**
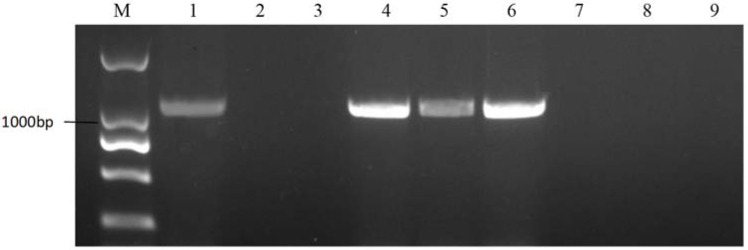
PCR amplification of KD4/KD7 in 18-512 sister lines. M represents the DNA Marker 2000; 1 represents the positive control; and 2 represents the negative control l. Lanes 3–9 are 18-512, 18-2169-1, 18-2169-2, 18-2169-3, 18-2170-1, 18-2170-2, and 18-2170-3.

**Figure 6 plants-11-01372-f006:**
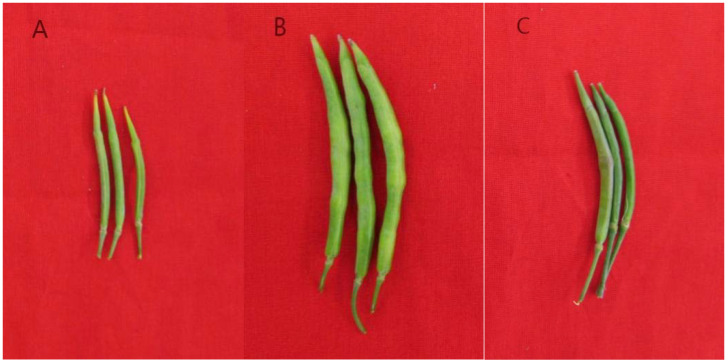
Pod enlargement of self-compatibility lines 18-2169 (**B**) and self-incompatibility lines 18-512 (**A**) and 18-2170 (**C**) in the open flower stage.

**Figure 7 plants-11-01372-f007:**
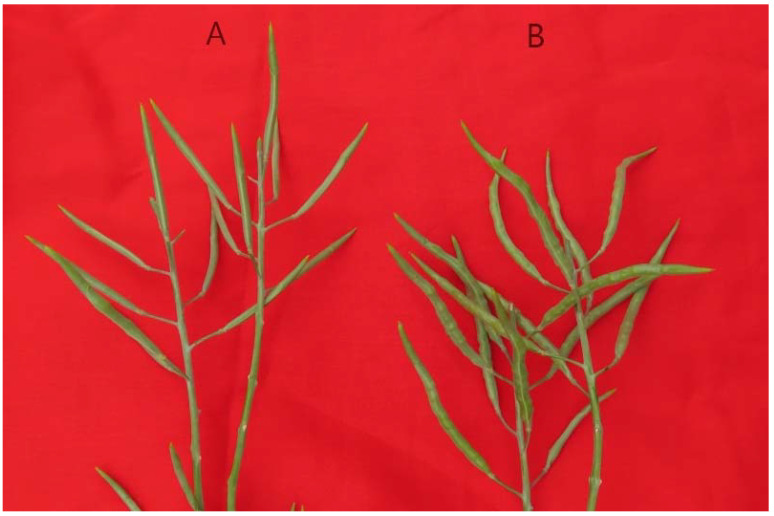
Pod enlargement results from hybridization of 18-2170 and 18-503 (**A**) and hybridization of 18-2169 and 18-503 (**B**) in the open flowering period.

**Figure 8 plants-11-01372-f008:**
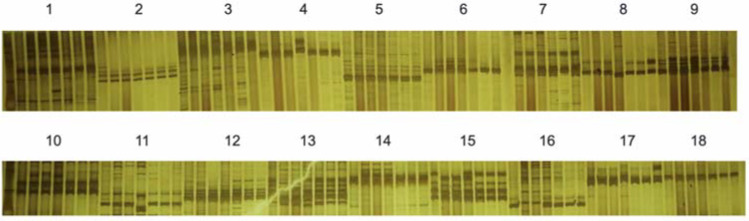
PCR amplification of nine pairs of SSR primers in 18-512 sister lines (18-512, 18-2169-1, 18-2169-2, 18-2169-3, 18-2170-1, 18-2170-2, 18-2170-3 from left to right). M represents the DNA Marker 500; numbers 1 to 18 represent SSR primers BoE188, BoE607, BoE162, BoE966, BoE222, BoE718, BoE002, BoE450, BoE882, BoE699, BoE379, BoE761, BoE723, BoE209, BoE875, BoE134, BoE734, BoE051.

**Figure 9 plants-11-01372-f009:**
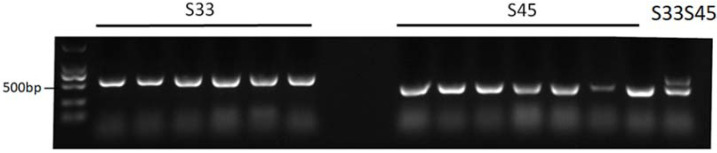
PCR amplification of two sets of primers (TF33-F/TF45-F/TS-R) in 14 cabbage materials (18-599, 18-622, 18-664, 18-665, 18-670, 18-684, 18-655, 18-659, 18-675, 18-679, 18-681, 18-695, 18-699, 18-651 from left to right).

**Figure 10 plants-11-01372-f010:**
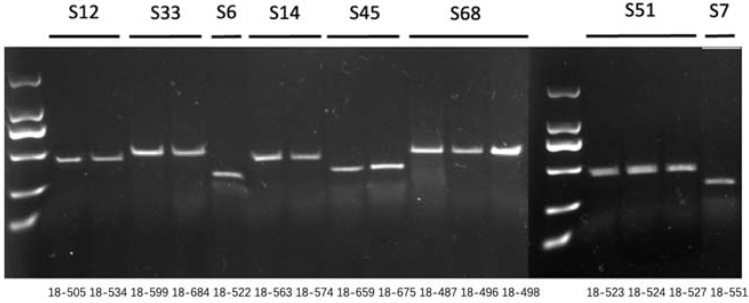
Image of agarose gel after the first digestion of 8 S-haplotypes.

**Figure 11 plants-11-01372-f011:**
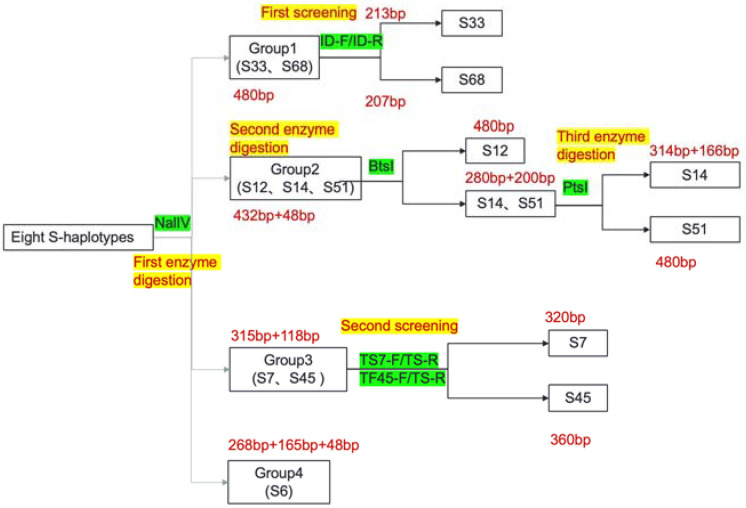
Identification flow chart of 8 S-haplotypes in cabbages.

**Figure 12 plants-11-01372-f012:**
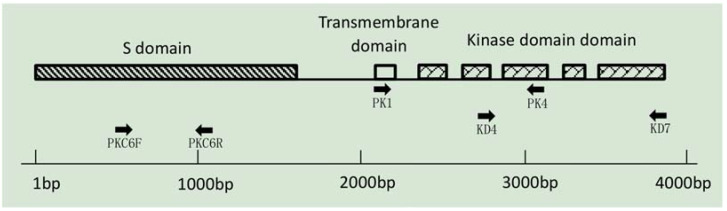
Schematic diagram of SRK gene structure and marker development position.

**Table 1 plants-11-01372-t001:** The matching degrees of primers with S-haplotype sequences.

S-Haplotype	GI Number in NCBI Database	PK1/PK4	PKC6F/PKC6R
S2-b	AB298890	-	79.07%
S4	M76647	95.56%	95.35%
S5	AB180898	-	74.42%
S6	AB054708	100.00%	95.35%
S7	AB180901	95.56%	95.35%
S8	AB024420	-	97.67%
S12	AB298891	93.33%	100.00%
S13	AB054710	95.56%	97.67%
S14	AB298892	97.78%	97.67%
S15	D85229	-	74.42%
S16	AB013720	-	97.67%
S17	AB054712 AB298894	97.78%	-
S22	AB054713 AB298895	-	97.67%
S23	AB054714 AB298897	95.56%	93.02%
S24	AB190355	88.89%	95.35%
S25	AB054715	95.56%	96.67%
S28	AB054716	95.56%	96.67%
S33	AB054718 AB298899	93.33%	93.02%
S35	AB054719	-	96.67%
S36	AB054720	-	93.02%
S39	AB298900	-	81.40%
S45	AB298901	-	100.00%
S50	AB054722	-	95.35%
S51	AB298903	-	100.00%
S52	AB054725	93.33%	81.40%
S57	AB298905	-	100.00%
S63	AB024416	95.56%	-
S64	Y18259	-	83.72%
S68	AB180903	88.89%	100.00%

**Table 2 plants-11-01372-t002:** Primers of class I S-haplotypes identification system.

Class	Primers	Sequences of Primers (5′-3′)	Tm/°C	Amplified Band
S7/S33	TS33-F	CCCCGAAAACTTCCGGAGTT	56	S7: 320 bp S33: 600 bp
	TS7-F	CGAACCACCAGTGCGATATGTACAG		
	TS-R	ACCATTCCGGATATCCGCATTT		
S33/S45	TF33-F	ATGGAATGAGGACTTTCCAATGCATC	56	S33: 573 bp S45: 364 bp
	TF45-F	CCCCGAACTCAATTCAATGGAACT		
	TS-R	ACCATTCCGGATATCCGCATTT		
S68/S33	ID-F	CATGTAAGTGCAGGTAGCTC	56	S68: 207 bp
	ID-R	CATGCATTTGAGCTTCCCACTA		S33: 213 bp

**Table 3 plants-11-01372-t003:** The specific primers of SRK.

Class	Primers	Sequences of Primers (5′-3′)	Tm/°C	References
Class I	PKC6F	CAATTTCACAGAGAATAGTGA	56	This study
	PKC6R	ACCATTCCGGATATCCGCATTT		
Class II	KD4	GAGGGCGAGAAGATCTTAATT	60	Park et al., 2002 [20]
	KD7	AAGACKATCATATTACCGAGC		

## Data Availability

Not applicable.

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
