# Peer review of "An Identification System Targeting the SRK Gene for Selecting S-Haplotypes and Self-Compatible Lines in Cabbage"

_plants, 2022, doi:10.3390/plants11101372_

Round 1

Reviewer 1 Report

The authors provided several primer sets developed based on SRK sequences to distinguish class I and class II S haplotypes and detect specific S haplotypes in cabbage breeding lines. Although these information may be important for self-incompatibility (SI) /compatibility (SC) breeding in cabbage, there are several concerns and points to be clarified.

1.The authors judged self-compatibility based on compatibility index. However a formula to calculate the compatibility index and the value of each line used were not shown. (Self-incompatibility: CI<1; self-compatibility: CI=>1) shown in line 319 means criterion to distinguish SI and SC. Also, how about the level of SC in each line?

2.Line195, SSR markers to evaluate the genetic background are not enough. At least 2 or 3 per chromosome are necessary.

3.In Table 3, the position of Tri-primers should be shown.

4. Line 213, how many F2 plants were investigated to show biased S haplotype segregation.

5. The authors found another line with class II haplotype among the 18-512 sister lines. It is wondering why the sister line include S15 haplotype. The pedigree of the sister lines should be shown.

6. Figure 11, in the text, S6 contained three bands, 268bp+165bp+48bp, but gel image indicated single band in S6. I do not understand.

Author Response

Reviewer 1

  1. 1. The authors judged self-compatibility based on compatibility index. However a formula to calculate the compatibility index and the value of each line used were not shown. (Self-incompatibility: CI<1; self-compatibility: CI=>1) shown in line 319 means criterion to distinguish SI and SC. Also, how about the level of SC in each line?

Response: Thanks for the comments. Calculation formulas have been added in the materials and methods section. The Table of field identification results (Supplementary Table 1) has shown the self-compatibility index of 58 inbred lines materials.

  1. Line195, SSR markers to evaluate the genetic background are not enough. At least 2 or 3 per chromosome are necessary.

Response: Thanks for the comments. Experiments were supplemented to ensure that each chromosome had at least two SSR primers. There was only one pair of SSR primers with high reliability on chromosome 9 of cabbage.

3.In Table 3, the position of Tri-primers should be shown.

Response: Thanks for the comments. The mark position was marked in line 197.

  1. Line 213, how many F2 plants were investigated to show biased S haplotype segregation.

Response: Thanks for the comments. It was showed in line 210-222.

  1. The authors found another line with class II haplotype among the 18-512 sister lines. It is wondering why the sister line include S15 haplotype. The pedigree of the sister lines should be shown.

Response: Thanks for the comments. We have added the pedigree of the sister lines in Supplementary Figure 1.

  1. Figure 11, in the text, S6 contained three bands, 268bp+165bp+48bp, but gel image indicated single band in S6. I do not understand.

Response: Thanks for the comments. The main band is 268bp, which was the feature band used to select S6. The size of other two bands is smaller, and then probably form smear, which can be visualized in the gel image.

Reviewer 2 Report

This manuscript described the results of development of a primer pair and identification system for distinguish Class I S-haplotype of SRK gene of the sporophytic self-incompatibility system. However, the prestation of the manuscript needs more logical and more concise.

1, the title needs more clearly and concise. For example, development of a primer pair and identification system for distinguish Class I S-haplotype of SRK gene in cabbage.

2, L79-88, the section “separation ratio of S-haplotypes deviates” is not related to this study, I think it should be deleted.

3, what is the flat-headed lines and round-headed lines, it needs a description.

4, L188-198, it needs more description on the heterosis and seed yield of new cross, but not only the success of pollination by honey bees.

5, L224-241, I suggest authors add a table for the results.

6, L96 delete 2.1. Subsection

7 please check the reference style.

Author Response

Reviewer 2

1, the title needs more clearly and concise. For example, development of a primer pair and identification system for distinguish Class I S-haplotype of SRK gene in cabbage.

Response: Thanks for the comments. We have adjusted the title based on the reviewer comments, “An Identification System Targeting the SRK Gene for Selecting S-haplotypes and Self-compatible Lines in Cabbage”. Our results included not only class I S-haplotypes identification, but also self-compatible lines selection.

2, L79-88, the section “separation ratio of S-haplotypes deviates” is not related to this study, I think it should be deleted.

Response: Thanks for the comments. This biased separation phenomenon of S-haplotype is discovered in this study, which can improve the importance of the molecular marker, because some S-haplotypes are easily to disappear. For example, the self-pollinated generations of cabbage hybrid '1186' (S7S33) and 'emerald' (S45S33) exists biased separation, after more than five generations of self-pollination, most inbred lines are S33, indicating that S33 is more advantageous to inherit to the offspring than S7 and S45. In order to enrich the diversity of S-haplotype in cabbage and reduce the loss of weak S-haplotype, Class I weak S-haplotype screening markers were developed. Therefore, it’s better to remain this part.

3, what is the flat-headed lines and round-headed lines, it needs a description.

Response: Thanks for the comments. Flat-headed lines and round-headed lines are two common head types. The ratio of width to height is close to 1:1 in round-headed lines, and the ratio of height to width is less than 0.8:1 in flat-headed lines. We have added the description in line 100-102.

4, L188-198, it needs more description on the heterosis and seed yield of new cross, but not only the success of pollination by honey bees.

Response: Thanks for the comments. It was found that there was no significant difference in head weight, height, width and biological yield when the self-compatible line 2169 or self-incompatible line 512 were crossed with the same female parent 18-503. We have added the description in line 192-194.

5, L224-241, I suggest authors add a table for the results.

Response: Thanks for the comments. We have added the flow chart (Figure 12).

6, L96 delete 2.1. Subsection

Response: Thanks for the comments. We have deleted it.

7 please check the reference style.

Response: Thanks for the comments. We have checked and modified it.

Round 2

Reviewer 1 Report

I have two minor points.

L254: self-pollinated is preferred instead of self-breeding.

L255: What does self-crossing mean? self-pollination?

Author Response

Reviewer 1

  1. L254: self-pollinated is preferred instead of self-breeding.

Response: Thanks for the comments. I have revised it according to comments.

  1. L255: What does self-crossing mean? self-pollination?

Response: Thanks for the comments. Self-crossing means self-pollination. In order to maintain the uniformity of wording, we have changed the self-crossing in the article into self-pollination.

Reviewer 2 Report

I have no future comments.

Author Response

Thanks for your review.